# Research on the concentration, potential and mission of science and technology innovation in China

**Ziyang Li[1]\*, Hongwei Shi[1], Hongda Liu[2,3]**

**1** School of management, Jiangsu University, Zhenjiang, China, **2** School of Economics & Management, Tongji University, Shanghai, China, **3** School of Management, Shanghai University, Shanghai, China

\* 3265096084@qq.com

**Data Availability Statement:** All relevant data are within the manuscript and its Supporting information files.

**Funding:** This project is supported by: National Social Science Foundation Educational Youth

## Abstract

Building an innovative country is a clear strategy for my country to promote economic, industrial, and social development. Mastering the status quo and late-comer advantages of technological innovation in my country's provinces is an important prerequisite for accurately positioning the innovation mission of each province. This article innovatively uses innovation concentration and innovation potential to measure and analyze the provincial scientific and technological innovation level at the inventory and incremental level. Taking the cross-sectional data of 31 provinces in my country in 2019 as a sample, construct the provincial "innovation potential-innovation concentration" evaluation The index system, through the entropy weight method, determines the weight of each evaluation index, and uses the TOPSIS method to conduct multi-objective evaluation of each province. According to the calculation results of provincial scientific and technological innovation concentration and innovation potential, an in-depth interpretation of the spatial distribution of my country's provincial scientific and technological innovation echelon is carried out. The study found that: the innovation concentration and innovation potential matrix echelon division of my country's provinces is divided into four echelons, namely high innovation. The echelon of drivers with high concentration and high innovation potential, the echelon of followers with high innovation concentration and low innovation potential, the echelon of dark horses with low innovation concentration and high innovation potential, and the echelon of latecomers with low innovation concentration and low innovation potential. Finally, the positioning strategy of the innovation mission of each echelon province is put forward to provide reference and enlightenment for the construction of innovative provinces and the promotion of a scientific and technological power.

## Introduction

Affected by the impact of the epidemic situation and the counter current of anti-globalization, the international economic environment is becoming increasingly complex, and the domestic innovation chain and industrial chain are also facing great challenges. General Secretary Xi

Project (CGA210242); The funders had no role in study design, data collection and analysis, decision to publish, or preparation of the manuscript.

**Competing interests:** The authors have declared that no competing interests exist.

Jin-ping stressed that our country should focus on building a new development pattern with "domestic circulation as the main body and domestic and international double circulation and mutual promotion". The "double cycle" development mode is a road of high quality development under the background of the "two big bureaus" of the great rejuvenation strategy of the Chinese nation and the great change of the world's hundred years. It is of great strategic and epochal significance"The new development pattern of "double circulation" puts forward new requirements for China's scientific and technological innovation, that is, we should regard the ability of scientific and technological originality as the power source of the birth of new technology, new economy and new business forms, and make it a powerful hand to optimize the production, distribution, circulation, consumption and other links, so as to fully guarantee the overall national security while promoting the economic and industrial transformation and development. Scientific and technological innovation enables industrial upgrading, which is conducive to the healthy development of new formats and new economy, promotes the rapid transformation of regional economy, and effectively solves the practical problems such as inadequate and unbalanced economic and social development. In the process of constructing science and technology innovation ecosystem, the objective endowment of innovation conditions plays a key role in the selection of development path, which is mainly divided into the innovation concentration of stock type conditions and the innovation potential of incremental type conditions. In terms of promoting regional economic development and accelerating domestic circulation, the innovation mission, as a navigation mark for the development of each province, is related to the planning matters such as the outline of innovation blueprint and the locking of innovation focus. Therefore, in-depth measurement of the concentration and potential of scientific and technological innovation in China's provinces and accurate positioning of the innovation mission of each province have become an important topic in the academic circle.

## Literature review

### Innovation concentration

The theoretical root of innovation concentration is the "structural hole theory" proposed by Ronald Burt (1992). He believes that in social networks, there is a state of discontinuity or no direct connection between individuals and the third party connecting the two has control and information advantages, which brings opportunities for improving social network concentration. The formal proposal of innovation concentration comes from a discussion on Zhou Qiren's [1] visit to Israel. He believes that the first premise of innovation is to gather the concentration of innovation power. Then, he stressed that the key to innovation concentration lies in innovation factor cluster. We should not only face up to the unbalanced geographical distribution of innovation phenomenon, but also pay attention to increasing innovation factor density and interaction frequency to enhance innovation concentration. Liu Gensheng [2] has a more concise definition of innovation concentration, that is, the aggregation degree of talents and innovation elements. In 2018, "innovation concentration" has become a high-frequency word of the two sessions in Shanghai. The political and business circles highly agree with the development logic of gathering innovation elements, enhancing innovation concentration, promoting cross-border integration, and catalyzing innovation achievements. Some scholars believe that the concentration of scientific and technological innovation will determine the competitiveness level of regional industrial economic development, and a new round of ranking competition of scientific and technological innovation strength has started. The importance of innovation concentration is self-evident, but there are few direct research results. For example, Xu Yingzhi [3] focuses on the research of provincial innovation concentration, and

takes Jiangsu Province as an example to put forward strategies to enhance innovation concentration, such as improving the level of innovation input, organization, output and environment, and creating a new version of innovation ecology. Xu Jun et al. [4] discussed the path to improve the innovation concentration of sponge City, while more scholars considered the innovation concentration into the innovation efficiency [5], innovation level [6] or innovation ability [7]. Based on the existing achievements, this paper believes that innovation concentration is an accurate measure of the actual level of regional scientific and technological innovation, which is convenient to understand the current situation of innovation, find out the "innovation stock" and provide reference for improving the aggregation degree of various innovation elements.

## Innovation potential

Academic research on innovation potential is mainly divided into the following aspects. One is to calculate the innovation potential. Wenke et al. [8] calculated the development potential of industrial innovation ecology, summarized the types of industrial innovation ecology, and found that the technology content of the industry has a positive impact on the development potential of industrial innovation ecology. The second is the evaluation of innovation potential. Lin Shanquan [9] and others established an evaluation model to evaluate the innovation potential of the National Independent Innovation Demonstration Zone in the Pearl River Delta. Izabela KRAWCZYK sokolowska established an evaluation system covering 36 influencing factors and systematically evaluated the innovation potential of Polish enterprises. Du š anMarkovi ć [10], which provides a reliable path to enhance the innovation capability of enterprises. Fang Weiwei et al. [11] established the evaluation system of industry university research collaborative innovation potential, and put forward suggestions and countermeasures. The third is the research on the influencing factors of innovation potential. Hu Jiya [12] explored the promotion effect of corporate profitability and organizational size on innovation potential of emerging industries, compared the difference between tax incentives and financial subsidies on innovation potential, and proposed the optimization path based on talent, policy, market and assessment. The fourth is to study the spatial agglomeration pattern of regional innovation potential. Wang Xingping et al. [13] discussed the relationship between innovation subject location and innovation potential. Wang Jiwu [14] has outlined the circular spatial distribution of Hangzhou's innovation potential. The research on the measurement of regional innovation potential, the selection of evaluation index and the spatial distribution of innovation potential lay the theoretical and methodological foundation for this paper.

## Innovation mission

According to the traditional economic theory, scientific and technological innovation is usually regarded as an exogenous variable of social economic model, and the government plays the role of market failure repairer. When scientific and technological innovation becomes the main power source of social economic cycle, innovation Mission Branch has obvious positive effects in stimulating market vitality, adjusting industrial structure, circulating social resources, and gathering innovative talents. Academic research on innovation mission is mainly reflected in the following categories: first, research on the importance of innovation mission. For example, Zhang Xuewen and others [15] point out that mission driven innovation is a major turn of theoretical innovation led by general secretary Xi Jinping's idea of building a community of human destiny, emphasizing on the leading role of the government in the implementation process of national innovation strategy. Chengran, et al. [16] took the UK as an example to illustrate the core position and leading role of national innovation institutions

in the innovation driven development process. Second, based on the macro perspective, it discusses the content changes of innovation mission in each period. For example, Liu Jielin et al. [17] combed and analyzed the new characteristics and new missions of China's regional innovation capability. Li Rui et al. [18] constructed the action framework of "top level strategy dimension core competence dimension basic support dimension" to build a world science and technology power. Wang Shuang [19] analyzed the evolution track of the National Independent Innovation Demonstration Zone, and said that the self-innovation zone will carry out adaptive exploration in the aspects of spatial distribution, personalized development, integration and collaboration, and the release of innovation factor potential. Based on the theory of integrated innovation and national innovation system, Chen Jin [20] proposed the connotation and framework of a new efficient, open and collaborative national innovation ecosystem. The third is to interpret the innovation mission with regional innovation path. Zhou Zhenjiang et al. [21] stressed that the construction of "Guangzhou Shenzhen Hong Kong Macao" scientific and technological innovation corridor will be the leading project to promote the coordinated development of "one nuclear, one belt and one zone" in Guangdong Province. YUAN Lanpin et al. [22] believe that the comparative advantage of regional innovation of the innovation system of the Yangtze River Economic Belt should be brought into play. Wu Zhengping et al. [23] summarized the evolution and laws of regional innovation system in Xinjiang Uygur Autonomous Region from four aspects of environmental factors, innovation subject, enterprise innovation and network degree, and put forward relevant innovation paths. Wang Fan et al. [24] believe that the five provinces in Northwest China should promote the intelligent and integrated development of industry by means of "empowerment, assignment and intelligence". Combined with the existing research, this paper formulates innovation mission positioning strategies for each innovation echelon province from the provincial perspective.

Innovation ability can reflect the ability of a city (region or country) to continuously nurture new global academic ideas, scientific discoveries, new technological inventions, and new industrial directions. It divides innovation capabilities into different dimensions, and analyzes the integration and transformation of different dimensions. Further landing the ability of innovation strategy, and then combining concepts and practices, can more efficiently promote innovation-driven development. The innovation ability pointed out in this article is a concept related to ability and motivation. It is not only the foundation and ability of the original innovation, but also the force that drives the development of innovation. It is a combination of the two meanings of innovation origin and innovation planning. Its function includes resource convergence, research and development. Four aspects: creation, achievement transformation and future leadership. Scientific research institutions are one of the main forces of original innovation in basic research and high-tech fields. Improving their original innovation capabilities is an inevitable requirement for enhancing the country's self-help innovation capabilities and building an innovative country. Combining the country as an innovation entity, through the integration of the country's existing innovation resources, improving the efficiency of scientific research personnel's innovation output, adding innovation results on the basis of the existing stock, and carrying out industry-university-research integration conversion, can better integrate the innovation strategy capabilities Turn the intangible into tangible, and promote the organic flow of scientific and technological innovation elements. Therefore, this research divides the outstanding national capabilities into: the inventory dimension (innovation concentration), the incremental dimension (innovation potential), and the mission dimension (leading the development of the industry).

Based on the above analysis, this study divides China's technological innovation into three dimensions: the stock dimension, the incremental dimension, and the efficiency dimension.

According to the actual force, late force and effective force to promote the development of innovation ability, it will give full play to its role to form China's scientific and technological innovation highland. However, there are different mechanisms of action within the three dimensions that coordinate with each other to promote development. According to the actual performance of innovation ability, we divide it into: innovation concentration, innovation potential and innovation mission. It is of great significance to explore the internal circulation mechanism of the resolution of various dimensions, and promote the high coordination and cooperation of the national science and technology innovation internal departments, organizational structure design, organizational policy implementation, and soft and hard environments, and provide ideas for the development of national innovation capabilities.

1. The mechanism of the actual force in the stock dimension
   The concentration of innovation is a quantitative expression of the atmosphere of the innovation environment, reflecting the country's investment in innovation. A high-density innovation atmosphere will enhance the enthusiasm of personnel, increase creativity and the acumen of grasping the frontiers of innovation. Being in an atmosphere of high innovation concentration will stimulate a high degree of integration of existing stocks to burst out stronger realism. The dimension of the national innovation stock reflects the country's existing heritage. The national scientific research department has a large number of high-quality innovative talents equipped with a rich software and hardware environment, and the practical power of innovation capabilities is fully reflected. The improvement of practical power is conducive to the formation of an innovative environment with a good atmosphere. The continuous optimization of the innovation environment further enhances the concentration of innovation. The increase of the concentration of innovation interprets the effect of reality, and finally forms: a relatively closed loop of innovation concentration—realistic power—good atmosphere.

2. The mechanism of late force in the incremental dimension
   Innovation potential, as the name suggests, is the unutilized force of the country in enhancing its innovation capability in the future, and it is an important factor in achieving the improvement of the country's innovation capability. The size of innovation potential directly determines the size of the increment of innovation capability, is an important driving force for the incremental dimension, and is also an important expression of the country's subsequent efforts. In the process of improving the innovation capability, the improvement of late force is conducive to the formation of promising innovation expectations, and the continuous optimization of innovation expectations further enhances the innovation potential, and finally forms a relatively closed loop, which interprets the effect of late force, and finally forms: Innovative potential—latency—good prospects are relatively closed loop.

3. The mechanism of effective force in the effectiveness dimension
   The mission of innovation is to measure the efficient conversion of innovation increments into stocks. High-efficiency conversion capabilities can more effectively convert increments into stocks, avoiding innovation redundancy and waste. In the dimension of innovation efficiency, the improvement of effective power is conducive to promoting the formation of high-quality innovation results, and the continuous optimization of innovation results further enhances the innovation mission, and finally forms a relatively closed loop.

Literature review shows that the measurement of the level of scientific and technological innovation is usually limited to the stock index or incremental index, and few studies integrate two perspectives. These two perspectives are actually different dimensions of the evaluation of

the level of regional scientific and technological innovation, which cannot be roughly confused. This paper innovatively uses innovation concentration and innovation potential to measure the stock and incremental level of provincial science and technology innovation level. Innovation concentration and innovation potential measure the level of regional scientific and technological innovation from the static and dynamic perspectives respectively, while innovation mission reveals the direction of regional innovation development. The dynamic and static of the three are combined, and the subjective and objective are unified. Under the new development pattern of "double cycle", it is of great significance to accurately measure the concentration and potential of provincial scientific and technological innovation, and define the provincial innovation mission accordingly, so as to stimulate the subjective initiative of innovation and development of innovation subjects in each chain within the jurisdiction and create an organic collaborative innovation ecosystem.

Science and technology innovation is complex system engineering. In the process of dynamic collaboration of many innovation subjects, all kinds of innovation objectives interact and influence each other. Therefore, in order to objectively and comprehensively measure the development level of provincial science and technology innovation, this paper uses Entropy TOPSIS method to determine the index weight in the evaluation index system of provincial innovation potential innovation concentration, and carries out multi-objective evaluation of each province's science and technology innovation level through TOPSIS method. According to the evaluation results, each province is divided into four echelons, namely high innovation concentration, high innovation concentration and high innovation concentration High innovation potential driver echelon, high innovation concentration and low innovation potential follower echelon, low innovation concentration and high innovation potential dark horse echelon and low innovation concentration and low innovation potential latecomer echelon. On the basis of comprehensive analysis of provincial innovation concentration and innovation potential, fully consider the characteristics of the new development pattern of "double cycle", more clearly describe the regional development path, and lead the innovation development with innovation mission, so as to inject new vitality into industrial upgrading and economic cycle.

## Construction of evaluation index system of innovation concentration and innovation potential

Since 1900, the research on regional innovation system has been increasing. Among them, Cooke [25] has made a landmark study on regional innovation system. He believes that regional innovation system is a regional organization system composed of geographically interrelated and clearly divided innovation subjects, such as enterprises, universities, scientific research institutions, etc. Asheim [26] and other scholars have studied the integration of regional innovation system and the development of regional industrial clusters. This idea has compacted the main line for the follow-up research, that is, the optimization of regional innovation system is essentially to enhance the regional industrial competitiveness and promote regional economic development. Gong Huang and Liu [27, 28] discussed the construction principles, organizational structure and strategic measures of regional innovation system from different perspectives. Yang et al. [29], Xie et al. [30] Based on the existing results, respectively evaluated the driving force of regional manufacturing innovation and regional innovation ecosystem. Theoretically speaking, the achievements have deepened the research on regional innovation system, which has positive significance. The first is to evaluate the regional innovation system from the spatial dimension, increase the research perspective of geographical distribution, and further expand the research boundary of national or provincial science and

technology innovation system. The second is to provide enlightenment for China to solve the practical problems of unbalanced innovation development and innovation resource aggregation and adjustment.

The basic structure of China's regional innovation system is affected by history, culture, economy, resources and other factors, showing more differences and characteristics in regional distribution. At the same time, under the background of "double cycle", the openness and innovation of China's economic development and industrial upgrading are increasing day by day. However, there are many problems in different provinces, such as industrial foundation, scientific and technological level and development speed the transformation efficiency is quite different. Therefore, the evaluation of provincial science and technology innovation level from the stock dimension (innovation concentration) and incremental dimension (innovation potential) of science and technology resources is helpful to fully grasp the current situation and late development advantages of China's Provincial Science and technology innovation, and provide an important reference for the accurate positioning of the innovation mission of each province.

## Establish the provincial innovation concentration innovation potential evaluation index system

This paper regards innovation concentration and innovation potential as two main aspects to measure the development level of provincial science and technology innovation, constructs a two-dimensional evaluation space of provincial science and technology innovation, and comprehensively evaluates the stock and incremental characteristics of provincial science and technology innovation. When constructing the corresponding evaluation index system, this paper comprehensively considers the following three factors: first, it emphasizes the multi-agent structure in the innovation system, that is, it emphasizes the interactive relationship of innovation subjects in the innovation ecology, such as universities, scientific research institutions, enterprises, governments, and intermediaries; The second is to emphasize the construction of provincial science and technology innovation chain, focusing on the generation, circulation and transformation of innovation elements in the innovation chain. The ability of innovation generation is reflected in the original innovation ability of universities and scientific research institutions, and the integration ability of innovation circulation is reflected in the ability of knowledge flow and technology transfer, The application ability of innovation transformation is shown in the ability of enterprises to transform technological innovation into market application; The third is to emphasize the interaction between innovation chain and industrial chain. Industrial chain creates a macro level social and economic environment for innovation chain, and provides a micro level industrial incubation capacity. Innovation chain feeds back on regional industrial chain in the form of economic benefits and employment growth. Under the guidance of the above principles, the evaluation index of innovation concentration selected in this paper is shown in Table 1, and the evaluation index of innovation potential is shown in Table 2. The indicators in each dimension involve six categories, namely: an innovation strategy source, B innovation integration, C innovation application, D innovation environment, e innovation incubation, f innovation performance.

The above six categories of indicators correspond to each endogenous link of the innovation chain (an innovation policy source, B innovation integration, C innovation application) and the exogenous link formed by interaction with the industrial chain (D innovation environment, e innovation incubation, f innovation performance), as shown in Fig 1. In terms of endogenous links, innovation strategy source describes the generation process of innovation elements, including innovation resource input, original innovation achievements, etc;

**Table 1. Indicators and weights related to innovation concentration.**

| Category | Indicator name | weights | Information entropy | Category | Indicator name | weights | Information entropy |
|---|---|---|---|---|---|---|---|
| A | Average research and experimental development full-time personnel equivalent per 10,000 people (person-year/10,000 people) | 0.0369 | 0.857 | C | Average number of effective invention patents per 10,000 industrial enterprises above designated size (pieces/10,000) | 0.0393 | 0.848 |
| A | The proportion of government R&D investment in GDP (%) | 0.0603 | 0.766 | C | Average external R&D expenditure of industrial enterprises above designated size (ten thousand yuan/unit) | 0.0302 | 0.883 |
| A | Number of invention patent applications accepted per 10,000 R&D personnel (pieces/10,000 people) | 0.0209 | 0.919 | C | Average expenditures for technological transformation of industrial enterprises above designated size (ten thousand yuan/unit) | 0.0123 | 0.952 |
| A | Number of invention patent applications generated by internal expenditures per 100 million yuan of R&D expenditure (pieces/100 million yuan) | 0.0206 | 0.920 | C | Proportion of the number of companies with e-commerce transaction activities in the total number of companies | 0.0172 | 0.933 |
| A | Number of authorized invention patents per 10,000 R&D personnel (pieces/10,000 people) | 0.0191 | 0.926 | C | Proportion of sales revenue of new products of industrial enterprises above designated size in sales revenue (%) | 0.0272 | 0.895 |
| A | Authorized teaching of invention patents generated by internal expenditures of R&D activities per 100 million yuan (pieces/100 million yuan) | 0.0110 | 0.957 | D | Internet penetration rate (%) | 0.0199 | 0.923 |
| A | The average number of domestic papers published per 100,000 R&D personnel (papers/100,000) | 0.0199 | 0.923 | D | Average number of entrepreneurial mentors per technology business incubator (person/person) | 0.0223 | 0.913 |
| A | Average number of international papers published per 100,000 R&D personnel (papers/100,000) | 0.0247 | 0.904 | D | The proportion of employees in the science and technology service industry in the tertiary industry (%) | 0.0269 | 0.896 |
| B | The number of scientific and technological papers in the same province and different units per 100,000 research and development personnel (papers/100,000) | 0.0266 | 0.897 | D | Education expenditure as a percentage of GDP (%) | 0.0268 | 0.896 |
| B | Number of scientific and technological papers from different provinces per 100,000 researchers and authors (papers/100,000) | 0.0182 | 0.930 | D | Proportion of tertiary education among the population aged 6 and over (%) | 0.0311 | 0.879 |
| B | Number of foreign scientific and technological papers per 100,000 R&D staff authors (papers/100,000) | 0.0133 | 0.948 | E | The average amount of loans obtained from financial institutions among the internal expenditures of R&D expenditures of industrial enterprises above designated size (ten thousand yuan/unit) | 0.0508 | 0.803 |
| B | Proportion of internal expenditures of R&D expenditures of universities and research institutes from enterprises (%) | 0.0183 | 0.929 | E | Intensity of venture capital investment of technology business incubator in the year | 0.0376 | 0.854 |
| B | Average transaction volume of technology market enterprises (according to flow direction) (10,000 yuan/item) | 0.0146 | 0.943 | E | Average amount of incubation base of each technology business incubator (ten thousand yuan/piece) | 0.0315 | 0.878 |
| B | The average expenditure for purchasing domestic technology by industrial enterprises above designated size (ten thousand yuan/item) | 0.0295 | 0.885 | E | The proportion of high-tech enterprises in the number of industrial enterprises above designated size (%) | 0.0166 | 0.936 |
| B | The average expenditure of technical introduction of industrial enterprises above designated size (ten thousand yuan/item) | 0.0803 | 0.688 | E | Average number of companies that graduated from each technology business incubator in the year (house/unit) | 0.0237 | 0.908 |

(*Continued*)

**Table 1.** (Continued)

| Category | Indicator name | weights | Information entropy | Category | Indicator name | weights | Information entropy |
|---|---|---|---|---|---|---|---|
| C | Proportion of R&D personnel among employed personnel in industrial enterprises above designated size (%) | 0.0150 | 0.942 | F | The ratio of the added value of the tertiary industry to GDP (%) | 0.0287 | 0.889 |
| C | Proportion of total internal expenditures of R&D activities of industrial enterprises above designated size in sales revenue (%) | 0.0112 | 0.957 | F | The ratio of main business income of high-tech industry to GDP (%) | 0.0106 | 0.959 |
| C | Proportion of enterprises with R&D institutions among the industrial enterprises above designated size in the total number of enterprises (%) | 0.0313 | 0.879 | F | The proportion of high-tech products exports to the total regional exports (%) | 0.0306 | 0.881 |
| C | Average number of invention patent applications per 10,000 R&D personnel of industrial enterprises above designated size (pieces/10,000 people) | 0.0193 | 0.925 | F | The proportion of the number of employees in the high-tech industry to the total number of employees (%) | 0.0257 | 0.900 |

Note: The corresponding relationship of category code is: A innovation source, B innovation integration, C innovation application, D innovation environment, E innovation incubation, F innovation performance; The original data sources are "China Regional Innovation Capability Evaluation Report", "China Science and Technology Statistical Yearbook", "China Science and Technology Paper Statistics and Analysis", "China Torch Statistical Yearbook", "China High-tech Industry Statistical Yearbook".

Innovation integration depicts the circulation process of innovation elements, including academic exchanges, technology transactions, etc; Innovation application depicts the transformation process of innovation elements, including R & D input, R & D output and benefit improvement. In terms of exogenous links, innovation environment describes the macro-economic environment of innovation chain, including hard environment and soft environment; Innovation Incubation depicts the micro growth space of innovation chain, including incubation strength and incubation quality; Innovation performance describes the economic spillover benefits of innovation chain, including regional economic performance and high-tech industry performance. These indicators reflect the characteristics of multiple interaction, whole chain connection, and co-prosperity of creation and production, which can comprehensively measure the provincial innovation concentration and innovation potential, and form a three-dimensional and multi angle portrait of regional scientific and technological innovation level.

## Determine the weight of each evaluation index by entropy weight method

When using TOPSIS method for multi-objective evaluation, determining the weight of evaluation index has a great influence on the final evaluation result. In order to reduce the subjective interference in the process of confirming the right, this paper uses the entropy weight method to determine the index weight in the process of TOPSIS evaluation, and uses the entropy weight TOPSIS method to calculate the innovation concentration and innovation potential of each province.

Entropy weight method belongs to the objective weighting method, which obtains the index weight with the help of information entropy. The information entropy of an indicator reflects the degree of variation of the indicator, which in turn reflects the amount of information contained in a given indicator. If the variation degree of the index is small, the information entropy is large, which will have a small impact in the evaluation process, so the weight is small. On the contrary, the weight is larger. The weights determined by the entropy weighting

**Table 2. Indicators and weights related to innovation potential.**

| Category | Indicator name | weights | Information entropy | Category | Indicator name | weights | Information entropy |
|---|---|---|---|---|---|---|---|
| A | Research and experimental development full-time personnel equivalent growth rate (%) | 0.0108 | 0.966 | C | The growth rate of expenditures for technological transformation of industrial enterprises above designated size (%) | 0.0058 | 0.982 |
| A | Government R&D investment growth rate (%) | 0.0145 | 0.954 | C | Growth rate of the number of companies with e-commerce transaction activities (%) | 0.0168 | 0.947 |
| A | Number of invention patent applications accepted (excluding enterprises) growth rate (%) | 0.0159 | 0.949 | C | Sales revenue growth rate of new products of industrial enterprises above designated size (%) | 0.0182 | 0.942 |
| A | The growth rate of the number of invention patents granted (%) | 0.0073 | 0.977 | D | Growth rate of Internet users (%) | 0.0198 | 0.937 |
| A | Growth rate of the number of domestic papers (%) | 0.1352 | 0.570 | D | Tech business incubator growth rate (%) | 0.0234 | 0.926 |
| A | Growth rate of the number of international papers (%) | 0.0463 | 0.853 | D | The growth rate of employees in the science and technology service industry (%) | 0.0058 | 0.981 |
| B | The growth rate of the number of scientific papers in different units in the same province (%) | 0.0624 | 0.802 | D | Growth rate of household consumption level (%) | 0.0133 | 0.958 |
| B | The growth rate of the number of scientific papers by authors in different provinces (%) | 0.0867 | 0.725 | D | Growth rate of education expenditure (%) | 0.0063 | 0.980 |
| B | The growth rate of the number of foreign scientific papers by authors (%) | 0.0126 | 0.960 | D | The population growth rate of the population of 6 years old and over with a tertiary education and above (%) | 0.0240 | 0.924 |
| B | The growth rate of corporate funds in the internal expenditures of R&D expenditures in universities and research institutes (%) | 0.0531 | 0.831 | E | Growth rate of loans obtained from financial institutions in the internal expenditures of R&D expenditures of industrial enterprises above designated size (%) | 0.0168 | 0.947 |
| B | Growth rate of technology market transaction amount (according to flow direction) (%) | 0.0086 | 0.973 | E | The growth rate of venture capital investment for technology business incubators that year (%) | 0.0644 | 0.795 |
| B | The growth rate of expenditures for domestic technology purchases by industrial enterprises above designated size (%) | 0.0438 | 0.861 | E | Growth rate of total technology business incubator incubation fund (%) | 0.0712 | 0.774 |
| B | The growth rate of expenditures for the introduction of technology by industrial enterprises above designated size (%) | 0.0500 | 0.841 | E | Growth rate of the number of high-tech companies (%) | 0.0073 | 0.977 |
| C | Growth rate of R&D personnel in industrial enterprises above designated size (%) | 0.0145 | 0.954 | E | Graduated companies of technology business incubator (%) | 0.0267 | 0.915 |
| C | Growth rate of total internal expenditure of R&D activities of industrial enterprises above designated size (%) | 0.0110 | 0.965 | F | Growth rate of the added value of the tertiary industry (%) | 0.0091 | 0.971 |
| C | The growth rate of the number of industrial enterprises above designated size with R&D institutions (%) | 0.0221 | 0.930 | F | Growth rate of main business income of high-tech industry (%) | 0.0188 | 0.940 |
| C | The growth rate of invention patent applications of industrial enterprises above designated size (%) | 0.0092 | 0.971 | F | High-tech product export growth rate (%) | 0.0184 | 0.942 |
| C | The growth rate of effective invention patents of industrial enterprises above designated size (%) | 0.0097 | 0.969 | F | Employment growth rate in high-tech industries (%) | 0.0122 | 0.961 |

(*Continued*)

**Table 2.** (Continued)

| Category | Indicator name | weights | Information entropy | Category | Indicator name | weights | Information entropy |
|---|---|---|---|---|---|---|---|
| C | Growth rate of external expenditure of R&D expenditure of industrial enterprises above designated size (%) | 0.0079 | 0.975 | | | | |

Note: The corresponding relationship of category code is: A innovation source, B innovation integration, C innovation application, D innovation environment, E innovation incubation, F innovation performance; The original data sources are "China Regional Innovation Capability Evaluation Report", "China Science and Technology Statistical Yearbook", "China Science and Technology Paper Statistics and Analysis", "China Torch Statistical Yearbook", "China High-tech Industry Statistical Yearbook".

method are generated from the data of each indicator and are objective and unique, avoiding the influence of subjective factors in the evaluation process.

Suppose that the original data of the second evaluation index in the second province is represented, then the steps of entropy weight method are as follows:

①.  Standardize the data:

$$m_{ij} = \frac{x_{ij} - \min(x_{ij})}{\max(x_{ij}) - \min(x_{ij})} \qquad (1 \le i \le m, 1 \le j \le n) \text{ (Positive Index)}$$

$$m_{ij} = \frac{\max(x_{ij}) - x_{ij}}{\max(x_{ij}) - \min(x_{ij})} \qquad (1 \le i \le m, 1 \le j \le n) \text{ (Negative Index)}$$

②.  Calculate the characteristic proportion of the $i$ province under the $j$ indicator $p_{ij}$:

$$p_{ij} = m_{ij} / \sum_{i=1}^{m} m_{ij}$$

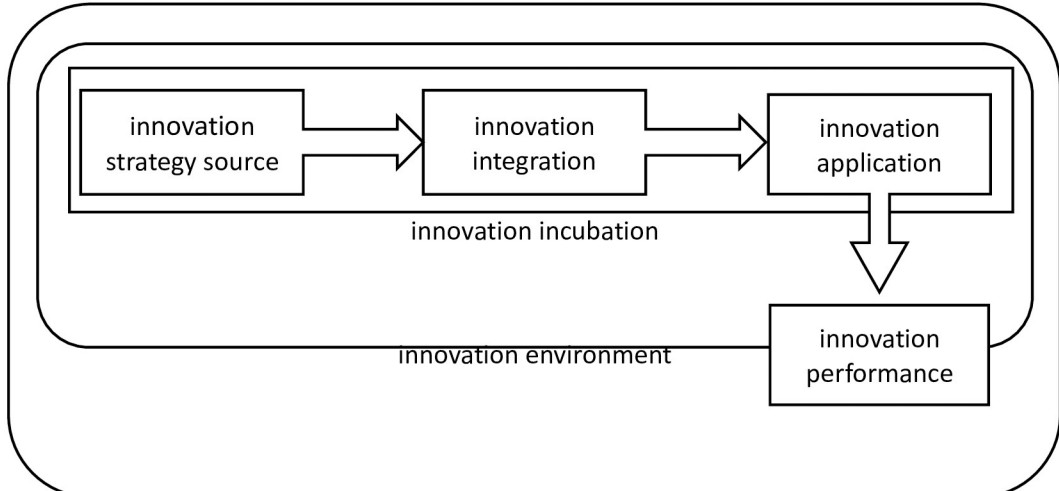

**Fig 1. Endogenous and exogenous links of innovation chain.**

③.  Computing information entropy $e_j$:

$$e_j = -\frac{1}{\ln m}\sum_{i=1}^{m}(p_{ij} \times \ln p_{ij})$$

④.  Calculate the weight of evaluation index $w_j$.

$$w_j = (1 - e_j)\bigg/\sum_{j=1}^{n}(1 - e_j)$$

After standardisation to remove the effect of the scale, the greater the degree of variation in the data contained in a given indicator, the greater the significance of that indicator in influencing the final result. The positive and negative ideal solutions in TOPSIS are the best and worst values of each indicator, respectively. Indicators with a large degree of variation will obviously have a greater impact when calculating the distance between a given solution and the ideal solution, and it is therefore necessary to assign a higher weight to them.

In the process of evaluating the innovation potential and innovation concentration of each province, the degree of variation of each evaluation index has a high explanatory validity for the contribution of each province to innovation, and can reflect the comparative advantage of each province in different dimensions of innovation. Therefore, this paper uses the entropy weighting method to assign weights to the corresponding indicator systems of innovation potential and innovation concentration in each province, so as to provide a basis for the next comprehensive evaluation of TOPSIS. The specific information entropy and weighting calculation results are shown in Tables 1 and 2 respectively.

## Multi objective evaluation of the concentration and potential of science and technology innovation in China's provinces

This paper selects TOPSIS method to comprehensively measure the innovation concentration and innovation potential of each province. TOPSIS method calculates the Euclidean distance of each evaluation object and two solutions on the basis of calculating the positive ideal solution and negative ideal solution of each evaluation index, and calculates the closeness degree between them and the positive ideal solution, and takes this as the basis of evaluation.

The steps of TOPSIS are as follows:

①.  The reverse index is forward processed, and the vector normalization matrix $Z$ is constructed:

$$x_{ij}^* = \max(x_{ij}) - x_{ij} \ (1 \le i \le m, 1 \le j \le n) \tag{1}$$

$$z_{ij} = \frac{x_{ij}^*}{\sqrt{\sum_{i=1}^{n}x_{ij}^{*2}}} \ (1 \le i \le m, 1 \le j \le n) \tag{2}$$

②.  The positive and negative ideal solutions corresponding to each evaluation index are determined.

$z_j^+$ and $z_j^-$ represent the maximum and minimum values of column $j$ in the normalized matrix respectively, then:

The positive ideal solution is:

$$\begin{aligned} Z^+ &= (\max\{z_{11}, z_{21}, \cdots, z_{m1}\}, \max\{z_{12}, z_{22}, \cdots, z_{m2}\}, \cdots, \max\{z_{1n}, z_{2n}, \cdots, z_{mn}\}) \\ &= (z_1^+, z_2^+, \cdots, z_n^+) \ (1 \leq i \leq m, 1 \leq j \leq n) \end{aligned}$$

The solution of negative ideal is:

$$\begin{aligned} Z^- &= (\min\{z_{11}, z_{21}, \cdots, z_{m1}\}, \min\{z_{12}, z_{22}, \cdots, z_{m2}\}, \cdots, \min\{z_{1n}, z_{2n}, \cdots, z_{mn}\}) \\ &= (z_1^-, z_2^-, \cdots, z_n^-) \ (1 \leq i \leq m, 1 \leq j \leq n) \end{aligned}$$

③. The Euclidean distances from provinces to positive and negative ideal solutions are calculated:

$$d_i^+ = \sqrt{\sum_{j=1}^{n} w_j (z_{ij} - z_j^+)^2} \ (1 \leq i \leq m, 1 \leq j \leq n)$$

$$d_i^- = \sqrt{\sum_{j=1}^{n} w_j (z_{ij} - z_j^-)^2} \ (1 \leq i \leq m, 1 \leq j \leq n)$$

Among them $w_j$ is the weight of the $j$ evaluation index, and the weight of each index is confirmed according to the entropy weight method.

④. Calculate the closeness of the comprehensive results of each province to the positive ideal solution:

$$S_i = \frac{d_i^-}{d_i^+ + d_i^-} \ (i = 1, 2, \cdots, m)$$

$0 \leq S_i \leq 1$, The closer to 1 for $S_i$, the better the performance of the evaluation object.

The positive ideal solution, negative ideal solution, closeness degree and corresponding ranking of scientific and technological innovation concentration and innovation potential of each province calculated by TOPSIS method are shown in Tables 3 and 4.

The Pearson correlation coefficient is -0.249, and the significance level is 0.088. It shows that there is no significant linear correlation between the provincial innovation level in the stock dimension and the incremental dimension. It is necessary to carry out two independent dimensions of research.

Therefore, combined with the results of Entropy TOPSIS, this paper takes the innovation concentration and innovation potential of each province as the coordinate axis, draws the scatter diagram according to the closeness degree, and divides the plane into four quadrants according to the average value of each province in the two dimensions, and the distribution result is roughly L-shaped. Because the data points are too dense, in order to facilitate observation, this paper takes the logarithm of the closeness degree and redraws the scatter diagram, as shown in Fig 2.

**Table 3. Ranking of provincial innovation concentration.**

| province | positive ideal solution | negative ideal solution | closeness degree | Ranking | province | positive ideal solution | negative ideal solution | closeness degree | Ranking |
|---|---|---|---|---|---|---|---|---|---|
| Beijing | 0.2010 | 0.4715 | 0.7011 | 1 | Hubei | 0.4930 | 0.1086 | 0.1805 | 17 |
| Shanghai | 0.3281 | 0.3782 | 0.5354 | 2 | Gansu | 0.5156 | 0.1106 | 0.1767 | 18 |
| Chongqing | 0.4297 | 0.2025 | 0.3203 | 3 | Xinjiang | 0.5445 | 0.1105 | 0.1687 | 19 |
| Guangdong | 0.4607 | 0.1943 | 0.2966 | 4 | Ningxia | 0.5242 | 0.1049 | 0.1667 | 20 |
| Hainan | 0.4928 | 0.1973 | 0.2859 | 5 | Shanxi | 0.5055 | 0.0981 | 0.1625 | 21 |
| Jiangsu | 0.4847 | 0.1694 | 0.2589 | 6 | Hunan | 0.5159 | 0.0980 | 0.1596 | 22 |
| Shaanxi | 0.4780 | 0.1613 | 0.2523 | 7 | Jiangxi | 0.5171 | 0.0967 | 0.1575 | 23 |
| Tianjin | 0.4632 | 0.1514 | 0.2463 | 8 | Henan | 0.5287 | 0.0988 | 0.1574 | 24 |
| Zhejiang | 0.5029 | 0.1487 | 0.2282 | 9 | Shandong | 0.5067 | 0.0935 | 0.1558 | 25 |
| Liaoning | 0.4843 | 0.1338 | 0.2165 | 10 | Fujian | 0.5055 | 0.0920 | 0.1540 | 26 |
| Guangxi | 0.5266 | 0.1423 | 0.2128 | 11 | Yunnan | 0.5210 | 0.0899 | 0.1472 | 27 |
| Sichuan | 0.4907 | 0.1292 | 0.2084 | 12 | Inner Mongolia | 0.5143 | 0.0875 | 0.1454 | 28 |
| Guizhou | 0.5031 | 0.1253 | 0.1993 | 13 | Jilin | 0.5252 | 0.0815 | 0.1343 | 29 |
| Heilongjiang | 0.5165 | 0.1237 | 0.1932 | 14 | Tibet | 0.5571 | 0.0703 | 0.1121 | 30 |
| Anhui | 0.5084 | 0.1185 | 0.1890 | 15 | Hebei | 0.5321 | 0.0624 | 0.1049 | 31 |
| Qinghai | 0.5373 | 0.1194 | 0.1819 | 16 | | | | | |

The high concentration high potential quadrant includes Hainan, Guangdong and Chongqing; The high concentration low potential quadrant includes Beijing, Shanghai, Tianjin, Shaanxi, Jiangsu and Zhejiang; The low concentration high potential quadrant includes Tibet, Qinghai, Gansu, Guizhou, Ningxia, Yunnan, Jiangxi and Jilin; The other provinces belong to the low concentration low potential quadrant.

**Table 4. Ranking of provincial innovation potential.**

| province | positive ideal solution | negative ideal solution | closeness degree | Ranking | province | positive ideal solution | negative ideal solution | closeness degree | Ranking |
|---|---|---|---|---|---|---|---|---|---|
| Tibet | 0.4310 | 0.6285 | 0.5932 | 1 | Shaanxi | 0.7064 | 0.1647 | 0.1891 | 17 |
| Hainan | 0.6529 | 0.3129 | 0.3239 | 2 | Hubei | 0.7053 | 0.1639 | 0.1886 | 18 |
| Qinghai | 0.6491 | 0.2883 | 0.3075 | 3 | Hunan | 0.7144 | 0.1641 | 0.1868 | 19 |
| Guizhou | 0.6619 | 0.2787 | 0.2963 | 4 | Fujian | 0.7084 | 0.1601 | 0.1844 | 20 |
| Ningxia | 0.6725 | 0.2657 | 0.2832 | 5 | Zhejiang | 0.7226 | 0.1626 | 0.1837 | 21 |
| Jiangxi | 0.6835 | 0.2569 | 0.2732 | 6 | Henan | 0.7142 | 0.1603 | 0.1833 | 22 |
| Yunnan | 0.6872 | 0.2510 | 0.2675 | 7 | Anhui | 0.7170 | 0.1607 | 0.1831 | 23 |
| Guangdong | 0.6896 | 0.2426 | 0.2602 | 8 | Guangxi | 0.7226 | 0.1594 | 0.1808 | 24 |
| Gansu | 0.7096 | 0.2367 | 0.2502 | 9 | Shandong | 0.7167 | 0.1481 | 0.1713 | 25 |
| Chongqing | 0.6978 | 0.2106 | 0.2318 | 10 | Beijing | 0.7235 | 0.1427 | 0.1647 | 26 |
| Xinjiang | 0.6919 | 0.1990 | 0.2234 | 11 | Shanghai | 0.7274 | 0.1388 | 0.1603 | 27 |
| Shanxi | 0.7004 | 0.1975 | 0.2199 | 12 | Jiangsu | 0.7382 | 0.1307 | 0.1504 | 28 |
| Hebei | 0.7117 | 0.1914 | 0.2120 | 13 | Heilongjiang | 0.7390 | 0.1273 | 0.1470 | 29 |
| Inner Mongolia | 0.7028 | 0.1741 | 0.1985 | 14 | Tianjin | 0.7312 | 0.1205 | 0.1415 | 30 |
| Jilin | 0.7166 | 0.1733 | 0.1948 | 15 | Liaoning | 0.7434 | 0.0989 | 0.1174 | 31 |
| Sichuan | 0.7085 | 0.1680 | 0.1917 | 16 | | | | | |

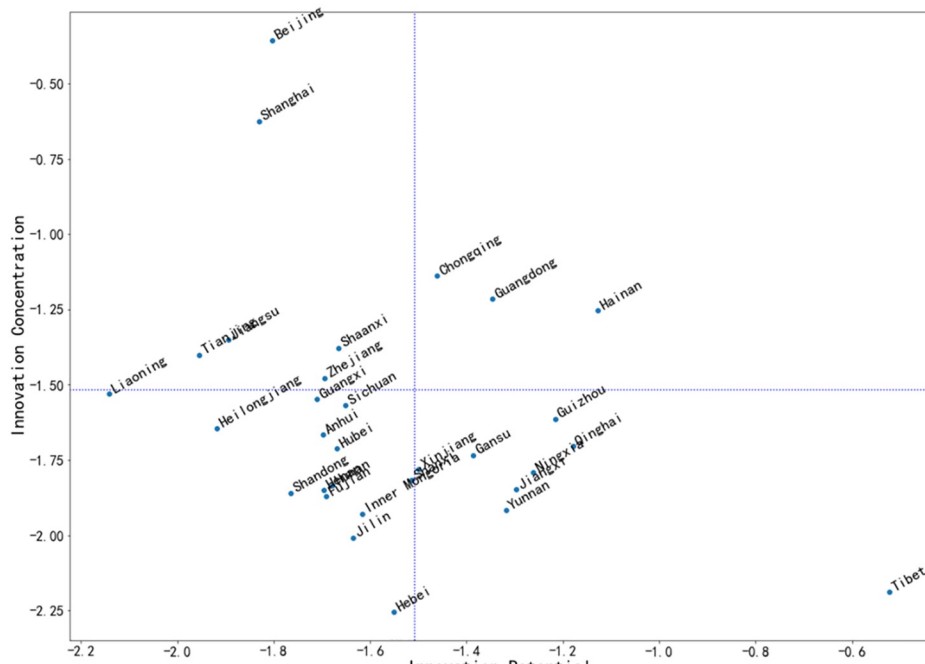

**Fig 2. Distribution of innovation potential and innovation concentration in different provinces (logarithm).**

## Stability analysis of evaluation results

In order to verify the stability of the results of the entropy-weighted TOPSIS method, this paper adjusts the weights of each indicator through the hierarchical analysis method and compares the differences in results with the entropy-weighted TOPSIS. The hierarchical analysis method is a combination of qualitative and quantitative decision analysis method, by judging the relative importance of each measurement indicator, and then obtain the weight of each indicator in the decision scheme. entropy weight method is a data-based weighting method, while the hierarchical analysis method is an empirical weighting method. The main steps are as follows:

1. Construction of indicator pairwise comparison matrix
   Since qualitative weighting is not immune to bias, Santy et al. propose the use of the consistency matrix method to construct a pairwise comparison matrix $A = [a_{ij}]_{m \times m}$. Where there are m evaluation indicators and $a_{ij}$ denotes the relative importance of indicator $i$ and indicator $j$, e.g. $a_{ij} = 1/3$ denotes the relative importance of indicator $i$ and indicator $j$ as 1:3. Each element $a_{ij} > 0$ and $a_{ij} = 1/a_{ij}$ of the pairwise comparison matrix A.

2. Calculate the eigenvalues and eigenvectors of the matrix
   The pairwise comparison matrix is first column normalized, i.e. each element is divided by the sum of the elements in the current column.

$$A \rightarrow A' = \begin{bmatrix} a_{11}/\sum_{i=1}^{m} a_{i1} & a_{12}/\sum_{i=1}^{m} a_{i2} & \cdots & a_{1m}/\sum_{i=1}^{m} a_{im} \\ a_{21}/\sum_{i=1}^{m} a_{i1} & a_{22}/\sum_{i=1}^{m} a_{i2} & \cdots & a_{2m}/\sum_{i=1}^{m} a_{im} \\ \vdots & \vdots & \ddots & \vdots \\ a_{m1}/\sum_{i=1}^{m} a_{i1} & a_{m2}/\sum_{i=1}^{m} a_{i2} & \cdots & a_{mm}/\sum_{i=1}^{m} a_{im} \end{bmatrix} = \begin{bmatrix} a'_{11} & a'_{12} & \cdots & a'_{1m} \\ a'_{21} & a'_{22} & \cdots & a'_{2m} \\ \vdots & \vdots & \ddots & \vdots \\ a'_{m1} & a'_{m2} & \cdots & a'_{mm} \end{bmatrix}$$

The normalized matrix is summed by row to obtain the eigenvectors for each row.

$$
A' \rightarrow v = \begin{bmatrix} \sum_{j=1}^{m}(a_{1j}/\sum_{i=1}^{m} a_{ij}) \\ \sum_{j=1}^{m}(a_{2j}/\sum_{i=1}^{m} a_{ij}) \\ \vdots \\ \sum_{j=1}^{m}(a_{2j}/\sum_{i=1}^{m} a_{ij}) \end{bmatrix} = \begin{bmatrix} \sum_{i=1}^{m} a'_{1i} \\ \sum_{i=1}^{m} a'_{2i} \\ \vdots \\ \sum_{i=1}^{m} a'_{mi} \end{bmatrix} = \begin{bmatrix} v'_1 \\ v'_2 \\ \vdots \\ v'_m \end{bmatrix}
$$

The feature vectors are normalized by column to obtain the indicator weights.

$$
v_A \rightarrow w = \begin{bmatrix} \dfrac{\sum_{j=1}^{m}(a_{1j}/\sum_{i=1}^{m} a_{ij})}{\sum_{k=1}^{m}(\sum_{j=1}^{m}(a_{kj}/\sum_{i=1}^{m} a_{ij}))} \\ \dfrac{\sum_{j=1}^{m}(a_{2j}/\sum_{i=1}^{m} a_{ij})}{\sum_{k=1}^{m}(\sum_{j=1}^{m}(a_{kj}/\sum_{i=1}^{m} a_{ij}))} \\ \vdots \\ \dfrac{\sum_{j=1}^{m}(a_{mj}/\sum_{i=1}^{m} a_{ij})}{\sum_{k=1}^{m}(\sum_{j=1}^{m}(a_{kj}/\sum_{i=1}^{m} a_{ij}))} \end{bmatrix} = \begin{bmatrix} \dfrac{v'_1}{\sum_{i=1}^{m} v'_i} \\ \dfrac{v'_2}{\sum_{i=1}^{m} v'_i} \\ \vdots \\ \dfrac{v'_3}{\sum_{i=1}^{m} v'_i} \end{bmatrix} = \begin{bmatrix} w_1 \\ w_2 \\ \vdots \\ w_m \end{bmatrix}
$$

Each column of the pairwise comparison matrix is multiplied by the element in the corresponding position in the indicator weights to obtain the $A''$ matrix.

$$
\begin{aligned}
A \rightarrow A'' &= \begin{bmatrix} a_{11} \cdot \dfrac{\sum_{j=1}^{m}(a_{ij}/\sum_{i=1}^{m} a_{ij})}{\sum_{k=1}^{m}(\sum_{j=1}^{m}(a_{kj}/\sum_{i=1}^{m} a_{ij}))} & \cdots & a_{1m} \cdot \dfrac{\sum_{j=1}^{m}(a_{ij}/\sum_{i=1}^{m} a_{ij})}{\sum_{k=1}^{m}(\sum_{j=1}^{m}(a_{kj}/\sum_{i=1}^{m} a_{ij}))} \\ \vdots & \ddots & \vdots \\ a_{m1} \cdot \dfrac{\sum_{j=1}^{m}(a_{ij}/\sum_{i=1}^{m} a_{ij})}{\sum_{k=1}^{m}(\sum_{j=1}^{m}(a_{kj}/\sum_{i=1}^{m} a_{ij}))} & \cdots & a_{mm} \cdot \dfrac{\sum_{j=1}^{m}(a_{ij}/\sum_{i=1}^{m} a_{ij})}{\sum_{k=1}^{m}(\sum_{j=1}^{m}(a_{kj}/\sum_{i=1}^{m} a_{ij}))} \end{bmatrix} \\
&= \begin{bmatrix} a_{11} \cdot w_1 & \cdots & a_{1m} \cdot w_m \\ \vdots & \ddots & \vdots \\ a_{m1} \cdot w_1 & \cdots & a_{mm} \cdot w_m \end{bmatrix}
\end{aligned}
$$

Summing the matrix $A''$ by rows to obtain the matrix $A'''$.

$$A'' \to A''' = \begin{bmatrix} \sum_{p=1}^{m} a_{1p} \cdot \dfrac{\sum_{j=1}^{m}(a_{pj}/\sum_{i=1}^{m} a_{ij})}{\sum_{k=1}^{m}\sum_{j=1}^{m}(a_{pj}/\sum_{i=1}^{m} a_{ij})} \\ \vdots \\ \sum_{p=1}^{m} a_{mp} \cdot \dfrac{\sum_{j=1}^{m}(a_{pj}/\sum_{i=1}^{m} a_{ij})}{\sum_{k=1}^{m}\sum_{j=1}^{m}(a_{pj}/\sum_{i=1}^{m} a_{ij})} \end{bmatrix} = \begin{bmatrix} \sum_{i=1}^{m} a_{1i} w_i \\ \vdots \\ \sum_{i=1}^{m} a_{mi} w_i \end{bmatrix}$$

Normalize $A'''$ and sum over all elements to obtain the eigenvalues.

$$\lambda = \sum_{t=1}^{m} \frac{\sum_{p=1}^{m} a_{tp} \cdot \dfrac{\sum_{j=1}^{m}(a_{pj}/\sum_{i=1}^{m} a_{ij})}{\sum_{k=1}^{m}(\sum_{j=1}^{m}(a_{kj}/\sum_{i=1}^{m} a_{ij}))}}{m \cdot \dfrac{\sum_{j=1}^{m}(a_{pj}/\sum_{i=1}^{m} a_{ij})}{\sum_{k=1}^{m}(\sum_{j=1}^{m}(a_{kj}/\sum_{i=1}^{m} a_{ij}))}} = \sum_{j=1}^{m} \frac{\sum_{i=1}^{m} a_{ji} w_i}{m \cdot w_i}$$

Consistency indicator $CI$ is defined as $CI = (\lambda - m)/(m - 1)$. $CI \to 0$ indicates that the better the consistency, i.e. the pairwise comparison matrix is well constructed. Consistency ratios $CR = CI/RI$, $CI$ are consistency indicators and $RI$ is a random consistency indicator. If $CI < 0$ then the degree of inconsistency of the matrix is within tolerance and its eigenvector can be used as the weight vector.

The distribution of innovation potential-innovation concentration obtained by the multi-objective evaluation of TOPSIS method after the assignment of weights through hierarchical analysis is shown in Fig 3. From the calculation results, it can be seen that the distribution of innovation potential-innovation concentration in each quadrant of each province remains basically the same under different weighting methods, which proves that the evaluation results of TOPSIS method have a certain degree of stability, and the positioning of the innovation status of each province has a high reliability.

## Four echelon provincial innovation mission positioning strategy

The driver echelon has high-quality innovation capital, followed by the black horse, laggard echelon provinces decreased in turn, the continuous accumulation of high-quality capital in the province will cause a lower rate of return on capital, prompting high-quality capital to overflow to the relatively capital deficient provinces, which can improve its rate of return again. Therefore, clarifying the short, medium and long-term innovation mission of each echelon Province, efficiently integrating the technology chain, talent chain and industry chain of each echelon Province, and promoting the innovation capital to flow from the driver echelon to the low-level echelon will help to solve the practical problems of uneven and insufficient economic, industrial and social development in China.

### Driver echelon

One belt, one road, the other is Guangdong, Hainan and Chongqing. Guangdong is benefited from the double blessing of the construction of the big bay area and the Pearl River Delta

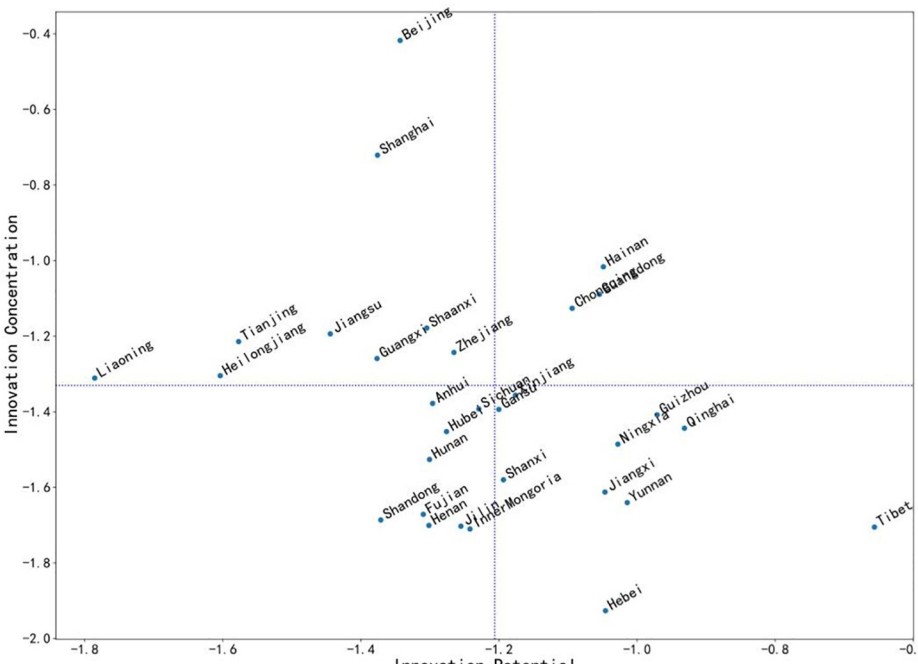

**Fig 3. Distribution of innovation potential—Innovation concentration by province (hierarchical analysis).**

economic belt. The free trade port construction brings Hainan development dividends, Chongqing is in the "one belt" and the Yangtze River economic belt, and it also takes the strategic dividend of western development. It is worth noting that after being diluted by the third and fourth tier cities in Guangdong, the innovation concentration in Guangdong is lower than that in Beijing, Shanghai and other regions. The future development of Hainan is promising, and there is still a large space for talent introduction, but the industrial foundation is weak and single, and the effect of policy implementation needs to be improved. Although Chongqing is in the driver echelon, its innovation potential is close to the boundary of the quadrant. As the innovation engine, the driver echelon needs to strengthen the basic innovation ability, limit breakthrough ability, format derivation ability and other innovation ability to further enhance the leading role.

1. Strengthen the ability of basic research and become the source of original innovation. Original innovation is a necessary condition for the driver echelon to maintain its innovation vitality. At present, the main body of scientific and technological innovation of the driver echelon is mainly enterprises, and the universities and scientific research institutions focusing on basic research are relatively weak. Therefore, it is necessary to speed up the construction of a number of first-class scientific and technological basic research centers, continue to promote the construction of high-level research universities, and develop emerging disciplines and interdisciplinary disciplines through multiple paths, Strive to achieve in-depth cooperation with science and technology enterprises, and promote the rapid landing of innovative ideas through a complete innovation chain.

2. Closely around the national innovation Bottleneck Breakthrough and key technology research, to provide more space and help. We should make great efforts to "create classics" and "break through the limit", avoid the accumulation of wealth with the help of capital

advantages in a single dimension, and transfer the tasks of technology dimensionality reduction, industrial transfer, talent return and digital empowerment to low-level echelon provinces in a planned way, Explain the innovation mission of the driver echelon.

3. We should actively build an innovative industrial ecology and create a new format of multi-agent linkage innovation. The innovation mission of the driver echelon is difficult to be realized by the innovation subject within the jurisdiction. The continuous blessing of policies, innovation atmosphere, residents' innovation literacy and various public resources can not be ignored. We should promote the efficient circulation and deep integration of various innovative elements in the innovation chain, promote the upgrading of "creativity" to "creation", continuously improve the industrial competitiveness, radiation driving force and business form derivative force, and always maintain the benign resonance between the innovation chain and the industrial chain.

## Follower echelon

The following echelon includes four municipalities directly under the central government of our country, as well as Jiangsu, Zhejiang and Shaanxi, which are strong provinces in the East and West. The following echelon has a high level of urbanization, a significant location advantage, abundant higher education resources, solid scientific research foundation, and good internal and external stock innovation conditions. But at the same time, these regions are facing three kinds of bottlenecks: first, the economic volume is too large, which leads to the path dependence of transformation and upgrading, and can release production capacity to adjust the innovation structure. Second, the innovation resources are concentrated but the circulation is blocked, and the innovation potential energy is enriched but the transformation degree needs to be improved. Third, as the convergence point of domestic and international circulation, the economic benefits brought by market opening are large, but the innovation benefits are insufficient, and there is external dependence. In view of the above problems, suggestions are as follows.

1. Improve the innovation radiation ability and become the innovation growth pole. Promote the diffusion of mature industries to the surrounding provinces, optimize their own industrial structure, and cultivate new innovation power points. We should not only pay attention to the division of industrial structure in the province, but also pay attention to the coordination of industrial system in the region, strive to turn from passive attraction to active guidance in the process of innovation elements aggregation, open up the innovation sharing channels with surrounding provinces, change the polarized one-way flow of resources into a win-win two-way interchange, and realize a wider range of economic cycle.

2. We should improve the circulation and transformation rate of scientific research achievements, and build collaborative innovation ecology of industry, University, research, government and application. It is necessary to improve the intellectual property protection system, scientific research incentive mechanism and technology trading market rules, open up the blocking points for the circulation of talents, information, technology, capital and other innovation elements, effectively coordinate the exchange of scientific and technological achievements among various innovation subjects, continuously optimize the allocation of innovation resources, avoid the accumulation and waste of innovation resources, and improve the smoothness of innovation chain with the help of innovation achievements transformation, Form a virtuous circle pattern of innovation chain and industrial chain.

3. We should improve the quality of urban opening up, fully integrate into the "double cycle", and pay attention to the technology spillover effect and learning effect brought by foreign direct investment and import and export trade"The new development pattern of "double circulation" needs high-quality opening to the outside world, improving the ability of independent innovation in the process of attracting foreign investment and introducing advanced technology and equipment, transforming the economic dividend brought by foreign trade into technical dividend, and finally into innovation dividend, so as to realize the double promotion of core scientific and technological competitiveness and independent innovation ability.

## Black horse echelon

The innovation foundation of black horse echelon is relatively weak, but the innovation growth rate is very prominent. This echelon province has similar structural problems: first, the scientific research strength is weak, and there is a lack of compound innovative talents. The second is the lack of supporting incubation capacity, and the innovation ecosystem needs to be improved. Third, the proportion of traditional industries is relatively high, and industries are concentrated in resource intensive areas, such as metal smelting, fine chemical industry, etc. However, in recent years, under the guidance of the national strategy, the development space, business environment and policy dividend of the black horse echelon have changed significantly, and the growth momentum has gradually emerged. In view of the above problems, suggestions are as follows.

1. Based on the province's own comparative advantages and innate endowment, the development path of differentiated innovation is planned. With the help of the implementation of national major scientific research projects, the construction of regional scientific and technological innovation chain will be promoted. For example, Gansu Province will take some military civilian integration industries as the prying point, introduce and build high-tech enterprises and scientific research institutions, and Guizhou will develop big data industry in combination with its own geographical advantages. On this basis, relying on the policy channel to build perfect supporting facilities, to promote the innovation growth power, to enable the local industrial economy, and to make use of the social multiple forces to deposit the innovation potential into the innovation strength.

2. Through the innovation growth path, build intelligent and comprehensive innovation demonstration Industrial Park. One belt, one road, is also being developed to attract talents in the field. We should build a multi domain, large integration, and strong innovation ecosystem, accelerate the construction of platforms for creating public spaces, incubators and accelerators, and gradually improve the logistics, finance and consulting services.

3. Adhere to the concept of ecological innovation and give consideration to the development of industrial economy and the construction of ecological civilization. We will promote the digital transformation of manufacturing industry and the upgrading of producer services, and transform resource intensive enterprises into technology intensive enterprises. One belt, one road, and the other, along the line of heavy industry, will be strictly controlled. The energy conservation and environmental protection standards must be controlled and the innovation and upgrading of the science and technology industry will be realized under the guidance of the "two mountain theory".

## Latecomer echelon

Most of China's northern and central provinces are in the latecomer echelon, and they are at a relative disadvantage in the aggregation process of innovation elements. There are three reasons: first, the layout of innovation system is scattered, and there is a lack of deep integration. Second, there is a lack of core support projects, and it is urgent to form a breakthrough innovation pole. Third, the level of regional coordination is not good, and the degree of urban integration needs to be improved. In order to improve the utilization rate of existing innovation resources and promote the leapfrog development of latecomer echelon, the suggestions are as follows.

1. We should cultivate multiple innovation subjects and promote a new situation of collaborative innovation. The development strategy is planned by the government, the breakthrough kinetic energy is released by the core enterprises, and the environmental support is constructed by the market and society, so as to fully activate and develop the innovation subject in stages, and lead the upgrading of various industries step by step. In addition, industry synergy drives innovation synergy, releases technology potential energy with the help of the brand influence of leading enterprises, promotes benign interaction of innovation elements, drives the surrounding small and medium-sized enterprises to open the road of digital and green development, continuously aggregates the regional industrial chain, and forms the overall industrial competitiveness.

2. We should create core innovation projects, cooperate with preferential policies, and tackle key problems in key areas. The industry structure of latecomer echelon is dominated by traditional industries, and lacks core projects as the driving force of scientific and technological innovation in the new round of industrial competition. In order to improve the overall level of innovation, we should introduce major scientific and technological projects with the help of preferential policies to stimulate the endogenous power of our own core innovation.

3. Create innovative metropolitan area and promote core cities to share knowledge with other regions in the province. We should further deepen the cultivation of advantageous fields, release the potential energy of scientific and technological innovation in provincial capitals and core cities, export technology and production capacity to the surrounding areas, promote the innovation achievements to become economic benefits, improve the content of innovation economy in the whole province, and form an innovation metropolitan area with multiple emerging innovation cities.

## Conclusion and outlook

Based on the evaluation and analysis of China's scientific and technological innovation in this article, the research proposes the following countermeasures and suggestions:

The first is to encourage school-site, school-enterprise cooperation, guide the main chain of the chain to move from formal cooperation to substantial cooperation, build a batch of in-depth cooperation, long-term and stable training and scientific research bases, promote the integration of major project projects and high-end talent projects, and realize advantageous resources Flow to key projects, key links and key talents to drive the increase in innovation concentration.

The second is to attract enterprises to provide colleges and universities with personnel, funds, and equipment support, to supplement the engineering practice links of the students, to

integrate the professional comprehensive capabilities, engineering practice capabilities, and scientific research and innovation capabilities of personnel, and to actively enhance the innovation reserve talents to achieve the growth of innovation potential.

The third is to cooperate with the digital technology incubator supervisory unit and the intelligent manufacturing enterprise incubator cultivation platform, obtain the support of venture capital institutions, expand the proportion of intelligent manufacturing technology investment in the overall investment quota, stimulate a new round of advanced manufacturing investment momentum, and cooperate for school-enterprise cooperation And smart manufacturing entrepreneurship provides new opportunities. Achieve the mission goal of innovation-driven national strategic growth.

In future research, researchers can further refine and adjust the evaluation index system. In addition, consider using a complete research framework to present the interactivity of indicators in various dimensions. With the supplement of these studies, a complete picture of China's scientific and technological innovation capabilities can be fully demonstrated.

## Supporting information

**S1 Data.**
(RAR)

## Author Contributions

**Data curation:** Hongda Liu.

**Writing – original draft:** Ziyang Li.

**Writing – review & editing:** Hongwei Shi.

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
