## [Decision Letter · Decision Letter 0]

23 Aug 2021

PONE-D-21-23193

Research on the concentration, potential and mission of science and technology innovation in China

PLOS ONE

Dear Dr. Liu,

Thank you for submitting your manuscript to PLOS ONE. After careful consideration, we feel that it has merit but does not fully meet PLOS ONE’s publication criteria as it currently stands. Therefore, we invite you to submit a revised version of the manuscript that addresses the points raised during the review process.

We look forward to receiving your revised manuscript.

Kind regards,

Mehdi Keshavarz-Ghorabaee

Academic Editor

PLOS ONE

Journal Requirements:

5. We note that Figure 3 in your submission contain map images which may be copyrighted. All PLOS content is published under the Creative Commons Attribution License (CC BY 4.0), which means that the manuscript, images, and Supporting Information files will be freely available online, and any third party is permitted to access, download, copy, distribute, and use these materials in any way, even commercially, with proper attribution. For these reasons, we cannot publish previously copyrighted maps or satellite images created using proprietary data, such as Google software (Google Maps, Street View, and Earth). For more information, see our copyright guidelines: http://journals.plos.org/plosone/s/licenses-and-copyright.

a) You may seek permission from the original copyright holder of Figure 3 to publish the content specifically under the CC BY 4.0 license.  

Reviewers' comments:

Reviewer's Responses to Questions

5. Review Comments to the Author

Reviewer #1: This paper is technically sound. In order to objectively and comprehensively measure the development level of provincial scientific and technological innovation, the author constructed the "Innovation Potential-Innovation Concentration" provincial evaluation index system, and used the entropy TOPSIS method to determine the index weights in the provincial innovation potential innovation concentration evaluation index system. Multi-objective evaluation of provincial innovation potential and innovation concentration has certain application value.

However, I don't know if it is a typesetting problem. The display of Table2 is confusing, and Figure2 is not very clear. In addition, when the TOPSIS method is used to evaluate the degree of innovation concentration, it is impossible to directly conclude that the highly concentrated provinces present the distribution characteristics of "multi-point dispersion and local aggregation".

The suggestion for this manuscript is minor revision, and I hope the author will integrate the analysis and evaluation of the innovation concentration in the paper more closely.

Reviewer #2: Dear Authors,

First, I would like to thank you for providing me with the opportunity to read your paper titled ‘Research on the concentration, potential and mission of science and technology innovation in China.’ I like your idea to draw attention to taking an accurate picture of scientific/technological innovation (STI) in China’s provinces in the context of its concentration, potential, and mission. The paper was well-written; the research topic is original; its structure is logical; the applied method for analysis is valid. My comments might hopefully help you to improve your paper.

1) The paper can be improved by providing a theoretical argument on why the concentration, potential, and mission of STI should be considered as its significant measurements. For this, you can use some theoretical explanations in the literature review.

2) In this section, you can also demonstrate several implications (or potential contributions) of this paper on academia and practitioners—maybe, policymakers—for potential readers’ understanding.

3) The literature review is well-written, using understandable terminology. In this section, it is required to resolve the inconsistency in citing references that should follow the journal guideline.

4) The analytical method is soundly technical. The journal recommends authors make all data underlying the findings fully available. Providing additional information on your data sources noted at the bottom of Table 1 (e.g., URL) would improve the transparency of your analysis and findings.

5) The results offer insights on ‘what is going on there.’ The categorization of provinces through the 2 x 2 matrix (Figure 2) is a rational approach. Based on the results, several valuable suggestions are drawn for the driver, follower, back horse, and latecomer echelons. This section can be improved by concluding with theoretical and/or practical implications of the findings.

I hope that the suggestions made here help enrich future versions of this paper.

Reviewer #3: This paper presents a study on the concentration, potential and mission of science and technology innovation in China. An approach based on the Technique for Order of Preference by Similarity to Ideal Solution (TOPSIS) and Entropy has been used in this study. I think that the paper is well-written and well-structured. However, I suggest that the authors consider the following comments to improve the paper:

1. I think the justification for using TOPSIS and Entropy should be explained more in the introduction section.

2. I think the paper should be improved by adding a new section or subsection to present other MCDM and weighting methods which can be used for the assessment process; like COPRAS (COmplex PRoportional Assessment), WASPAS (Weighted Aggregates Sum Product Assessment), SECA (Simultaneous Evaluation of Criteria and Alternatives), MEREC (MEthod based on the Removal Effects of Criteria) and EDAS (Evaluation based on Distance from Average Solution).

3. A table should be added to the literature review section to show the main features of the previous studies and the current study.

4. Acronyms and abbreviations should be defined at their first mention. Please check them in the manuscript. For example: Technique for Order of Preference by Similarity to Ideal Solution (TOPSIS)

5. The equations should be numbered and organized.

6. Section headings should be descriptive and concise.

7. I think a sensitivity analysis should be made based on changing criteria weights to show the stability of results.

8. A conclusion section should be added to present final discussion, concluding remarks and suggestions for future research.

Overall, I think the paper needs to be revised before publication.

---

## [Author Response · Author response to Decision Letter 0]

2 Sep 2021

Response to Reviewer 1 Comments

Dear Reviewer:

First of all, thank you very much for your feedback! These are excellent suggestions, and we wholeheartedly agree with your points. Here are our responses in terms of your feedback:

Point 1:

However, I don't know if it is a typesetting problem. The display of Table2 is confusing, and Figure2 is not very clear. In addition, when the TOPSIS method is used to evaluate the degree of innovation concentration, it is impossible to directly conclude that the highly concentrated provinces present the distribution characteristics of "multi-point dispersion and local aggregation".

Response 1: 

Thank you so much for your comments. We adjusted the format and deleted the map part based on the comments of the editorial department. The description of the relevant distribution characteristics has also been revised.

* Again, we thank you for your helpful inputs! We believe the new considerations and major revisions should better show the paper's rationale.

Take care, and Thank you so much.

Response to Reviewer 2 Comments

Dear Reviewer:

First of all, thank you very much for your feedback! These are excellent suggestions, and we wholeheartedly agree with your points. Here are our responses in terms of your feedback:

Point 1:

The paper can be improved by providing a theoretical argument on why the concentration, potential, and mission of STI should be considered as its significant measurements. For this, you can use some theoretical explanations in the literature review.

Response 1: 

Thank you very much for your suggestions. We have added the concepts and elicited meanings of these dimensions. Thank you！

Point 2:

In this section, you can also demonstrate several implications (or potential contributions) of this paper on academia and practitioners—maybe, policymakers—for potential readers’ understanding.

Response 2: 

Thank you for your careful knowledge. We supplemented the contribution of this research, highlighting its potential impact on policy.

Point 3:

The literature review is well-written, using understandable terminology. In this section, it is required to resolve the inconsistency in citing references that should follow the journal guideline.

Response 3: 

Thanks for your guidance. We have re-adjusted the document format in conjunction with the journal guidelines.

Point 4:

The analytical method is soundly technical. The journal recommends authors make all data underlying the findings fully available. Providing additional information on your data sources noted at the bottom of Table 1 (e.g., URL) would improve the transparency of your analysis and findings.

Response 4: 

Thank you for your suggestion, we have uploaded the source file and code as an attachment.

Point 5:

The results offer insights on ‘what is going on there.’ The categorization of provinces through the 2 x 2 matrix (Figure 2) is a rational approach. Based on the results, several valuable suggestions are drawn for the driver, follower, back horse, and latecomer echelons. This section can be improved by concluding with theoretical and/or practical implications of the findings.

Response 5: 

Thank you very much for your suggestion. We have re-added the relevant discussion. Thanks for your comments

* Again, we thank you for your helpful inputs! We believe the new considerations and major revisions should better show the paper's rationale.

Take care, and Thank you so much.

Response to Reviewer 3 Comments

Dear Reviewer:

First of all, thank you very much for your feedback! These are excellent suggestions, and we wholeheartedly agree with your points. Here are our responses in terms of your feedback:

Point 1:

I think the justification for using TOPSIS and Entropy should be explained more in the introduction section.

Response 1: 

Thank you very much for your suggestions. We have supplemented the related theories and analysis of Topsis and entropy, thank you for your contribution.

Point 2:

I think the paper should be improved by adding a new section or subsection to present other MCDM and weighting methods which can be used for the assessment process; like COPRAS (COmplex PRoportional Assessment), WASPAS (Weighted Aggregates Sum Product Assessment), SECA (Simultaneous Evaluation of Criteria and Alternatives), MEREC (MEthod based on the Removal Effects of Criteria) and EDAS (Evaluation based on Distance from Average Solution).

Response 2: 

Thank you very much for your help with our article. We have re-added the source of weights and analysis methods, combined with your comments to strengthen the interpretation of the evaluation process. Finally, we emphasize the suitability and uniqueness of this research method. thanks for your help

Point 3:

Acronyms and abbreviations should be defined at their first mention. Please check them in the manuscript. For example: Technique for Order of Preference by Similarity to Ideal Solution (TOPSIS). The equations should be numbered and organized. I think a sensitivity analysis should be made based on changing criteria weights to show the stability of results.

Response 3: 

Thanks for your guidance. We adjusted the corresponding content one by one to make the article structure more appropriate.

Point 4:

I think a sensitivity analysis should be made based on changing criteria weights to show the stability of results. A conclusion section should be added to present final discussion, concluding remarks and suggestions for future research.

Response 4: 

We re-use the analytic hierarchy process to get new weights. Combining the comparison of the two weights and conducting sensitivity analysis, it is found that the location changes of the provinces have not changed, which shows the stability and validity of the results. At the same time, we have added the conclusions and prospects to make research more valuable.

* Again, we thank you for your helpful inputs! We believe the new considerations and major revisions should better show the paper's rationale.

Take care, and Thank you so much.

---

## [Editor Report · Decision Letter 1]

6 Sep 2021

Research on the concentration, potential and mission of science and technology innovation in China

PONE-D-21-23193R1

Dear Dr. Liu,

We’re pleased to inform you that your manuscript has been judged scientifically suitable for publication and will be formally accepted for publication once it meets all outstanding technical requirements.

Kind regards,

Mehdi Keshavarz-Ghorabaee

Academic Editor

PLOS ONE

---

## [Editor Report · Acceptance letter]

15 Sep 2021

PONE-D-21-23193R1 

Research on the concentration, potential and mission of science and technology innovation in China 

Dear Dr. Liu:

I'm pleased to inform you that your manuscript has been deemed suitable for publication in PLOS ONE. Congratulations! Your manuscript is now with our production department. 

Kind regards, 

on behalf of

Dr. Mehdi Keshavarz-Ghorabaee 

Academic Editor

PLOS ONE